# Structural Insights into Endostatin–Heparan Sulfate Interactions Using Modeling Approaches

**DOI:** 10.3390/molecules29174040

**Published:** 2024-08-26

**Authors:** Urszula Uciechowska-Kaczmarzyk, Martin Frank, Sergey A. Samsonov, Martyna Maszota-Zieleniak

**Affiliations:** 1Laboratory of Molecular Modeling, Department of Theoretical Chemistry, Faculty of Chemistry, University of Gdansk, Wita Stwosza 63, 80-308 Gdańsk, Poland; u.uciechowska@gmail.com (U.U.-K.); sergey.samsonov@ug.edu.pl (S.A.S.); 2Biognos AB, P.O. Box 8963, 40274 Göteborg, Sweden; martin.frank@biognos.se

**Keywords:** glycosaminoglycans, endostatin, protein–glycosaminoglycan interactions, molecular docking, repulsive scaling–replica exchange molecular dynamics

## Abstract

Glycosaminoglycans (GAGs) play a key role in a variety of biological processes in the extracellular matrix (ECM) via interactions with their protein targets. Due to their high flexibility, periodicity and electrostatics-driven interactions, GAG-containing complexes are very challenging to characterize both experimentally and in silico. In this study, we, for the first time, systematically analyzed the interactions of endostatin, a proteolytic fragment of collagen XVIII known to be anti-angiogenic and anti-tumoral, with heparin (HP) and representative heparan sulfate (HS) oligosaccharides of various lengths, sequences and sulfation patterns. We first used conventional molecular docking and a docking approach based on a repulsive scaling–replica exchange molecular dynamics technique, as well as unbiased molecular dynamic simulations, to obtain dynamically stable GAG binding poses. Then, the corresponding free energies of binding were calculated and the amino acid residues that contribute the most to GAG binding were identified. We also investigated the potential influence of Zn^2+^ on endostatin–HP complexes using computational approaches. These data provide new atomistic details of the molecular mechanism of HP’s binding to endostatin, which will contribute to a better understanding of its interplay with proteoglycans at the cell surface and in the extracellular matrix.

## 1. Introduction

Glycosaminoglycans (GAGs), a class of long linear anionic polysaccharides with disaccharide repetitive units [1], play important roles in many crucial biological processes through their interactions with numerous protein partners [2,3,4]. GAGs are known to interact with intracellular, secreted or membrane-bound proteins and thereby regulate cell proliferation, migration, differentiation, angiogenesis, axon guidance, responses to central nervous system (CNS) injury, virus attachment and entry, and immune response [5,6,7]. Therefore, a detailed structural understanding of protein–GAG interactions is important for deciphering the biological mechanisms they are involved in and for designing new therapeutic strategies for diseases such as cancer and autoimmune disorders [3]. Despite the importance of protein–GAG interactions, a limited number (less than 100) of experimental 3D structures of protein–GAG complexes are available [8,9]. Most of them represent complexes with heparin (HP) with a degree of polymerization (dp) ranging from dp2 to dp7 [8,10,11]. This relatively low number of available experimental structures is due to the structural complexity, conformational heterogeneity and flexibility of GAGs and to the difficulty of obtaining oligosaccharides of a defined length and sulfation pattern in sufficient amounts for structural studies [6]. Furthermore, experimental techniques are often not sufficient for studying the interactions occurring at an atomistic level in detail, whereas their combination with computational approaches has proved to be useful for the detailed characterization of structure–function relationships in protein–GAG complexes [12,13,14,15]. This study focuses on a GAG-binding protein, endostatin, which is an anti-angiogenic proteolytic fragment of collagen XVIII [16] that interacts with several proteins regulating angiogenesis at the surface of endothelial cells including VEGFR1 and VEGFR2 [17]. Recombinant endostatin has been tested in clinical studies and significantly improves the survival of patients with late-stage non-small-cell lung carcinoma. Clinical data have also been collected on patients with other cancers treated by endostatin [18]. Endostatin’s binding to heparan sulfate (HS) is enhanced in presence of Zn^2+^, and its antiproliferative effect on endothelial cells is stimulated by fibroblast growth factor-2 (FGF2) [19]. Mutations of amino acid residues that coordinate Zn^2+^ lead to the decrease of endostatin anti-tumor activity [20], but not its anti-proliferative effect on endothelial cells, stimulated by vascular endothelial growth factor (VEGF) [21], or its anti-angiogenic activity in a model of angiogenesis induced by VEGF [19]. The regulation of endostatin activities by Zn^2+^ appears to be context-dependent and warrants further investigation since Zn^2+^ itself is able to bind HP [22]. Zinc-dependent dimers have been observed in crystals of human endostatin [23], but attempts to co-crystallize murine endostatin with an HP dp4 failed [24]. The crystal structures of both human and murine endostatin are available, and two HP binding sites have been identified by mutagenesis [25] and predicted by molecular docking [19].

In this study, we aimed to improve our understanding of the molecular recognition of HP and HS by endostatin in its monomeric form. For this, we applied several computational approaches including conventional molecular docking and a novel effective docking approach based on a repulsive scaling replica exchange molecular dynamics technique [26] that is particularly efficient for GAGs [27] and allows for the docking of long GAG molecules. The obtained binding poses were further analyzed using molecular dynamics (MD) approaches. We analyzed the impact of individual endostatin residues on HP/HS binding in terms of their free energy contributions to GAG binding depending on the GAG type, length and sulfation pattern. Furthermore, we in silico investigated the effect of Zn^2+^ on the formation of endostatin–HP complexes. The effect of mutation of several amino acid residues of endostatin was computationally predicted to contribute to HP/HS binding to these GAGs. Our results provide new insights into the structural properties of endostatin–GAG complexes. They could also be used to fine-tune the affinity of endostatin to HP/HS oligosaccharides of various lengths and sequences and so to modulate the biological activities of endostatin mediated by HP/HS, which could be potentially an attractive therapeutic target.

## 2. Results

### 2.1. Conventional Molecular Docking and MD Simulations

Autodock 3 (AD3) molecular docking runs followed by MD simulations were used to predict and analyze the recognition of different GAGs by endostatin in its monomeric form. The length of the docked GAGs was varied from dp2 to dp8, aiming to define the core GAG binding site and to understand the dependence of binding on the GAG length. Since there are in total 48 possible different combinations of HP/HS dp2 periodic units [9], we selected several of the most representative combinations, including HP and four HS variants, for this analysis. Although a systematic screening study of all 48 combinations would be interesting to perform, this was, however, beyond the scope of this work due to the absence of available reference experimental data. The dp8 was used as the longest GAG due to the limitation of the AD3’s performance quality by the number of the ligand’s degrees of freedom [28]. All GAGs docked to the region of the protein corresponding to the extent of its positive electrostatic potential (Figure 1A). For HP structures with a ^1^C_4_ ring puckering conformation of IdoA(2S) (dp2, dp4, dp6 and dp8), two overlapping clusters of docking solutions were observed. They were all located in the region close to or between the two α-helices (α1, α2), which is the most positively charged area of endostatin (Figure 1). The docking solutions for dp2 and dp4 were close to the α2 helix (Figure 1B). The free energy of binding (ΔG) is more favorable for longer HP oligosaccharides. The residues that contributed most favorably to ΔG were Arg47, Arg53, Arg62, Arg63 and Arg66 for HP dp2 and Arg27 and Arg139 for HP dp4 (Table 1, Appendix A). All docking solutions for ^1^C_4_ HP dp6 and dp8 were located between the α1 and α2 helices (Figure 1), with a major contribution of Arg62, Arg63, Arg66, Arg24 and Arg27 (Table 1, Appendix A).

All docked HP with a ^2^S_0_ ring puckering conformation of IdoA(2S) bound at least partially similarly to HP, with a ^1^C_4_ conformation for IdoA(2S) (Figure 2). The only significant differences were observed for the binding poses of the HP dp4 and dp6. For HP dp4, the structures were aligned with the α1 helix and directed towards the N-terminal loop of endostatin (Figure 2), whereas they were located near α1 helix for the ^1^C_4_ HP dp4 (Figure 1). For HP dp6, in case of a ^2^S_0_ ring puckering conformation, an additional second most populated cluster was obtained, in which the structures can be seen as an elongation of the ones in the most populated cluster presented for oligosaccharides with both ring puckering conformations. The absence of similar differences for HP dp2 can be explained by the fact that dp4 may be the minimal length which adopts the same binding pose as longer polysaccharides. On the other hand, dp6 and dp8 seem to be long enough to interact with the α1 and α2 helices and N-terminal part of the protein at the same time. This is in agreement with our previous studies on the important role of IdoA(2S) ring conformational flexibility in HP for docking and MD-based free energy calculations in protein–GAG systems [29]. In addition to the different binding modes obtained for the HP molecules with different IdoA(2S) ring conformations, the corresponding free energies of binding, which were used as a measure of the stability of the obtained complexes, were also different (Table 1), suggesting a potential multipose binding characteristic for other protein–GAG systems of these short oligosaccharides [30,31,32] and a potential dependence of their affinity on the IdoA(2A) ring pucker. 

The docking solutions obtained for desulfated HS dp2, dp4, dp6 and dp8 formed clusters that overlapped so that the corresponding binding poses of longer oligosaccharides represented the elongation of shorter ones (Appendix A). The free energy of binding for the desulfated HS was significantly less favorable than that for HP, confirming the importance of electrostatic interactions established by sulfate groups (Table 1, Appendix A). The binding pose of the desulfated HS dp2 was closer to the α1 helix compared to HP ^1^C_4_ and ^2^S_0_ dp2 (Appendix A). The desulfated HS dp4 has a binding pose similar to that of HP ^1^C_4_ dp4, with both interacting with the α2 helix and being directed to the N-terminal loop of endostatin (Appendix A). For desulfated HS dp6 and dp8, a higher similarity of their poses can be observed in comparison to both HP ^1^C_4_ and HP ^2^S_0_ (Figure 3), suggesting a more defined binding pose.

Docking predictions for HS dp8 with the repeating units GlcNS(6S)-GlcA, GlcNS-GlcA and GlcNS-IdoA(2S) and with IdoA(2S) in ^1^C_4_ and ^2^S_0_ ring conformations yielded two clusters (Figure 4A–D). Docking solutions corresponded to similar binding affinities and were located between the α1 and α2 helix. The binding affinity of HS dp8 with the GlcNS(6S)-GlcA unit, GlcNS-GlcA and GlcNS-IdoA(2S) ^2^S_0_ yielded on average ΔG ~ −50 kcal/mol, whereas the HS dp8 with the GlcNS-IdoA(2S) ^1^C_4_ unit resulted in ΔG ~ −60 kcal/mol (Table 1). This might be due to the stronger contribution of the Arg24, Arg27 and Arg139 residues to the binding (Appendix A). For HS (GlcNS-IdoA(2S))_4_, depending on the Ido(2S) ring pucker conformation, the second cluster was localized essentially differently. 

To summarize the results obtained for HS oligosaccharides with the conventional docking approach, it was observed that not only the net sulfation degree but the particular sulfation pattern and IdoA(2S) can affect the affinity of these molecular complexes. At the same time, the performed analysis does not allow us to distinguish a particular binding pose but rather a co-existence of binding poses corresponding to a similar binding free energy (multipose binding). 

Since Zn^2+^ binding is critical for endostatin to maintain a stable native structure [33] and is required for its biological activities, we also conducted molecular docking followed by an MD analysis of HP and several HS dp8 ligands using endostatin’s structure and including a Zn^2+^ ion (PDB ID: 1BNL) to analyze if the effect of the Zn^2+^ is also detectable with our computational approach (Appendix A). The reason to use dp8 in this part of the study was that while it is the biggest size of GAG that AD3 still can effectively deal with [28], it was shown to bind endostatin in a previous experimental study [19]. All the docked poses were found in the previously described binding sites predicted in the absence of Zn^2+^. In case of HP, the binding affinity obtained in our calculations increased in the presence of a Zn^2+^ ion (Table 1 and Table 2), while HP was bound closer to the α1 helix when the ion was present (Figure 5). Although there was no direct contact between HP dp8 and the ion observed, the more favorable binding energy could be attributed to a long-range electrostatics effect. The solutions obtained for HS with the repeating units GlcNS(6S)-GlcA and GlcNS-GlcA also formed clusters similar to those obtained in absence of Zn^2+^. In addition, the cluster located near the N-terminus of endostatin and Zn^2+^ was found for both HS oligosaccharides. For HS, this comprised GlcNS-IdoA(2S) disaccharide units with a ^1^C_4_ conformation; the structures within the cluster were further extended into the binding region than for the same dp6 with a ^2^S_0_ conformation (Appendix A). The presence of Zn^2+^ did not significantly affect the binding free energies in the complexes of endostatin with HS dp8 in contrast to with HP dp8 (Table 1 and Table 2, Appendix A).

### 2.2. RS-REMD Analysis

As we stated before, oligosaccharide longer than dp8 are too long to expect a reliable performance of conventional docking approaches [34]. To check the interactions between endostatin and longer oligosaccharides, we performed RS-REMD simulations for the endostatin–HP dp24 complex. This length was chosen with the aim of being able to cover all the positive patches on the protein surface at once, while the HP sequence was used for this proof-of-concept analysis since this sequence is usually the most representative in the PDB and can be clearly defined in computational protein–GAG analyses. The convergence time of this complex during the RS-REMD simulation was around 5 ns. Then, the HP remained in the same binding site for the next 55 ns and was aligned to the axis of the protein and located close to the α2 helix, which is very similar to the results obtained for shorter GAGs (Figure 6).

The trajectory obtained from RS-REMD simulation was analyzed using the MM-GBSA approach. According to our earlier studies [27], the most favorable values of electrostatic energy in vacuo obtained from an MM-GBSA analysis correspond to the most likely structures in protein–GAG complexes. Therefore, a refinement procedure was performed for the twenty structures with the lowest values of this binding free energy component. The starting structures and the structures obtained as a result of the refinement are presented in Appendix A.

The 20 endostatin–HP dp24 complex structures were sorted and numbered from the most (1) to the least (20) electrostatically favorable. In 15 out of the 20 refined structures, the oligosaccharide adopted a binding pose, in which it was extended and parallel to the axis of the protein, with visible contacts between the HP dp24 and the N-terminal loop of the endostatin (Appendix A), which is similar to the results from AD3. However, in complexes 2, 3, 4, 7 and 18, HP dp24 bound mainly to the central part of the protein. The MM-GBSA analysis identified complexes 2 and 17 as the most energetically favorable (Table 3). Although they represented essentially different structures, in both of them HP dp24 was in contact with the N-terminus of endostatin (Figure 7).

The analysis of the averaged decomposed binding energy of the endostatin amino acid residues presented in Table 4 did not suggest that the residues located within the α1 and α2 helices of the protein are mainly responsible for the stabilization of the complex structure with HP dp24, as they were in the case of the shorter HP/HS oligosaccharides. The Arg27 and Arg139 residues had the lowest ΔG values (−12.9 and −11.2 kcal/mol, respectively) (Table 4), which is consistent with previous reports [19]. Other favorable binding contributions were found mostly for Arg24, and two other residues previously predicted to contribute to HP binding [19], Arg53 and Arg63. The decomposition analysis was performed in detail for the two most energetically favorable complexes (Appendix A). The amino acids Arg24, Arg27, Arg63 and Arg64, which positively contributed to the binding of HS, are located within the Walker A motif of endostatin (Figure 8), which consists of the RX_(2-3)_R sequence interacting with ATP [35]. This suggests that HS could modulate the ATPase activity of endostatin, in as much as HP has been reported to bind to ATPases [36,37].

For both best-scoring binding poses, an analysis of the hydrogen bonds formed between endostatin and HP dp24 atoms was performed using CAT software [38] (Appendix A). The largest population of hydrogen bonds involved the Arg27 and Arg53 amino acid residues. Additionally, for pose 1, the Arg63 residue was involved, while for pose 2 the Arg24 and Arg139 residues were the hydrogen bond donors with the largest population. These results are consistent with the results obtained by the per residue decomposition analysis. The high number of the multiple stabilizing hydrogen bonds between the protein and HP dp24, as well as the electrostatic contribution, suggest that there is rather low specificity in GAGs’ recognition by endostatin.

An analysis of the of HP dp24 ring puckering conformation of the best-scoring binding poses was also performed. The MD trajectories obtained in the refinement procedures were analyzed using CAT software [38]. Appendix A shows the conformational changes occurring during MD for individual HP dp24 rings for the two best-scoring complexes. For both complexes, there was a ^1^C_4_ ring puckering conformation of IdoA(2S) clearly dominating for most of the rings during the entire MD simulation time, with the exception of the IdoA(2S) in the fifth position in the sequence, when counted from the non-reducing end, for which boat/twist conformations were also present. The dominance of the ^1^C_4_ ring puckering conformation was previously suggested by both experimental and theoretical studies [39]. The boat/twist conformation was more populated for the highest scoring complex, where it was observed for most of the analyzed frames. This may be explained by the fact that this oligosaccharide residue strongly interacts with Arg63, which potentially favors the boat/twist conformation.

### 2.3. Unbiased MD Simulation Analysis

Unbiased MD simulations with different versions of the force field parameters in comparison to AMBER MD simulations were performed in NAMD in order to complement the findings obtained with AMBER. The protocol parameters were optimized for the unbiased MD simulations in NAMD independently of the ones in the AMBER MD simulations. The studied molecular system consists of endostatin and HP dp5 (Figure 9). The use of an HP probe of this particular length, different by a single saccharide unit from the dp4 and dp6 used in conventional docking, was supposed to provide even more heterogeneity in the conditions in the performed simulations to allow for more general and less approach-biased conclusions about the structural properties of the analyzed complexes. In total, 35 MD simulations were performed, starting from five different spatial positions (A-E) of HP dp5 (Figure 9A). All the obtained structures of the endostatin–HP dp5 complexes (Figure 9B) represented a combination of binding poses with a most commonly obtained one that was located near or between the α1 and α2 helix, which is in agreement with the results described in the previous sections (see Figure 1 and Figure 2 for the most representative cluster, cluster 1). Therefore, despite the structural heterogeneity of the obtained binding poses of HP oligosaccharides and the differences in the protocols used in the conventional docking and unbiased MD simulations, one common representative cluster of HP binding poses was calculated by both approaches.

Individual simulations covered timescale of 0.5 μs; however, some were extended to 1 μs (Figure 10). As can be seen from the accumulated atom–atom contact trajectory, the binding of HP dp5 typically occurred in less than 50 ns. The full-residue atom–atom contact trajectory provided a detailed view of the dynamics of the interaction of HP dp5 with endostatin during unbiased MD simulations. The binding modes sampled were quite diverse; however, certain binding motives did re-occur independently of the starting structure. Interactions occurred mainly between HP dp5 and the Arg24, Arg27, Arg53, Arg63 and Arg139 residues. Although the interactions obtained by different replicas of the MD were reproducible, the ligand still could have become stuck in ‘artificial sites’. It should be noted that the accumulated timescale (20 μs) sampled here exceeds the timescales typically sampled in MD simulations reported in the scientific literature. 

### 2.4. Endostatin Mutations

The aspartate residues of endostatin located close to the predicted GAG binding region unfavorably contributed to the binding of HP and HS oligosaccharides. Therefore, we performed single mutations of Asp30, Asp56 and Asp65; a double mutation of Asp30 and Asp56; and a triple mutation of Asp30, Asp56 and Asp65 to arginine to determine whether these mutations could improve the binding of endostatin to HP and affect the binding pose. The mutations significantly affected the MM-GBSA free energy, rendering it more favorable, from −53.8 to −64.1, −79.3 and −81.6 kcal/mol for the single D65R, D30R and D56R mutant, respectively (Table 5). For the double and triple mutants, the energy was −107.2 and −111.4 kcal/mol, respectively. Asp30 and Asp56 might thus contribute more unfavorably to the binding to HP and HS oligosaccharides than Asp65 in the wild-type protein. 

## 3. Discussion

In this work, a systematic and rigorous computational analysis of endostatin–HP/HS complexes was performed to obtain novel insights into their structural and energetic properties depending on the GAG length, sulfation pattern and the presence of Zn^2+^. Several complementary computational approaches were applied to this system. For short GAGs (dp2-dp8), Arg24, Arg27, Arg62, Arg63, Arg66 and Arg139 are the residues which contribute the most to the stabilityof the complex. For a longer ligand, HP dp24, an RS-REMD simulation with an endostatin monomer as a receptor identified two representative binding poses with the most favorable binding energies. For the first pose, the most favorable amino acid residues are located close to the N-terminal loop (His1, Asp4) and the α2 helix (Arg62, Arg63 and Arg66). The His1 residue interacts with HP dp24 and this interaction potentially induces substantial conformational changes in the endostatin N-terminus. For the second pose, Arg residues from the α2 helix are mainly involved in the complex’s stabilization (Arg24, Arg27, Arg53, Arg63 and Arg139). Several of these residues belong to the Walker A motif and so potentially could compete with ATP for binding to this endostatin region, affecting its catalytic activity. This is in agreement with the fact that the plasma membrane Ca^2+^-ATPase is inhibited by HP [37]. Our results show that different HP/HS oligosaccharides are docked to similar binding sites on the endostatin surface independently of the docking method applied. GAG poses are mainly located in the region between the two α-helices (α1, α2), which is in agreement with the electrostatic potential calculations data obtained by the PBSA approach. The MM-GBSA analysis shows that the binding of desulfated HS dp2-dp8 is significantly less favorable than the binding of sulfated HP dp2-dp8, suggesting the key role of electrostatic interactions in this system. The influence of three endostatin residues unfavorably contributing to GAG binding was further analyzed. MD studies of endostatin mutants complexed with HP showed that the D30R and D56R mutations render the binding energy more favorable. We also performed a comprehensive analysis of HP and HS dp8 structures using an endostatin structure including a Zn^2+^ ion. The presence of Zn^2+^ essentially affected the binding free energies in the complexes, rendering them comparable to the ones seen in the absence of Zn^2+^ for HP dp8, but not for HS dp8, while for HS this yielded an additional docking pose located near the N-terminus of endostatin and Zn^2+^.

Our in silico approaches allowed us to improve our understanding of the molecular mechanisms underlying endostatin’s binding to HP and HS to generate models of several endostatin–HP/HS complexes and to identify the specific amino acid residues stabilizing these complexes. The results obtained using various theoretical and experimental techniques are consistent with each other and with previous experimental data [19,25].

## 4. Materials and Methods

### 4.1. Endostatin’s Structure

The crystal structure of human endostatin (PDB ID: 1BNL 2.9 Å) was used for the molecular docking and MD simulations. The amino acid residues of endostatin were numbered starting with the first residue of endostatin (His1572 in the full-length collagen XVIII α1 chain according to the UNIPROT numeration). Mutated structures were produced by the Chimera UCSF program [40].

### 4.2. Electrostatic Potential Calculations

Electrostatic potential isosurfaces were calculated using the PBSA program from AmberTools 20 [41], with a grid spacing of 1 Å.

### 4.3. Conventional Molecular Docking

The conventional docking of the flexible ligands HP/HS dp2, dp4, dp6 and dp8 to the endostatin receptor was performed with Autodock 3 (AD3) [42], with a grid spacing of 0.375 Å. The following ligands were used: 1. HP (disaccharide unit GlcNS(6S)-IdoA(2S) in both IdoA(2S) ^1^C_4_ and ^2^S_0_ ring puckering conformations); 2. desulfated HS (disaccharide unit GlcNAc-IdoA); 3. HS (disaccharide unit GlcNS(6S)-GlcA); 4. HS (disaccharide unit GlcNS-GlcA); 5. HS (disaccharide unit GlcNS-IdoA(2S) in both IdoA(2S) ^1^C_4_ and ^2^S_0_ ring puckering conformations). The ^1^C_4_ and ^2^S_0_ ring puckering conformations for all IdoA(2S) residues within the same molecule were defined by the dehidral angles (C1, C2, C3, C4) and (C1, O5, C5, C4), with their values corresponding to the X-ray structure of HP (PDB ID: 1HPN), where the HP molecule is available in both conformations [43]. For the desulfated HS, only IdoA ^1^C_4_ was considered since it is energetically a more favorable pucker conformation [39]. The Lamarckian genetic algorithm was used, with an initial population size of 300, 10^5^ generations, 9995 × 10^5^ energy evaluations and 10^3^ independent runs. The fifty top docking solutions were clustered by the DBSCAN algorithm [44], with a neighborhood search radius of 3.0–3.2 Å and a minimal number of cluster members of three. The cluster numbering used throughout the manuscript corresponds to their size ranking, with cluster 1 to be the most populated, cluster 2 the second populated, etc. In the clustering procedure, the distance between two structures was defined as the root mean square of their atomic distances while pairing up the nearest atoms of the same type, which takes into account the fact that GAGs are made up of periodic units [45]. The docking studies of endostatin with Zn^2+^ were performed using the parameters available from the AMBER 16 software package [41]. 

### 4.4. Molecular Dynamics and MM-GBSA Calculations

MD simulations of endostatin in a complex with HP and HS oligosaccharides were carried out in AMBER 16 [41]. A complex was solvated in a truncated octahedron TIP3P water periodic box with a 10 Å distance from solute atoms in each direction to the box wall and neutralized by Na^+^ or Cl^−^ counterions. The ff99SBonlysc [46] and GLYCAM06 force field [47] parameters were used for the endostatin and GAGs, respectively. MD simulations were preceded by two energy minimization steps: 1500 cycles of the steepest descent and 1000 cycles of a conjugate gradient with harmonic force restraints of 10 kcal/(mol·Å^2^) on solute atoms, and then 3000 cycles of the steepest descent and 3000 cycles of a conjugate gradient without restraints. The system was then heated from 0 to 300 K for 10 ps with harmonic force restraints of 10 kcal/(mol·Å^2^) on the solute atoms, and a 100 ps MD equilibration was run at 300 K and 10^5^ Pa in NTP without restraints. A cut-off of 8 Å was applied to treat non-bonded interactions, and the Particle Mesh Ewald method was applied to treat long-range electrostatic interactions. Following the equilibration procedure, 25 ns of productive MD runs were performed within periodic boundary conditions in NTP. 25 ns is of a sufficient length for the MD simulation of stable protein–GAG complexes [11]. To furthermore increase the sampling, three random initial structures from each cluster were simulated, which resulted in 75 ns of simulations for each cluster. The results from each of three simulations were independently analyzed. The energetic post-processing of the trajectories and per residue energy decomposition were carried out using MM-GBSA with igb = 2 for the protein–GAG complexes for all the frames obtained from the productive MD run. We did not consider explicitly entropic contributions to the binding, since taking into account entropy would increase the overall uncertainty in the calculated binding energies [48,49]. Therefore, the calculated free energies should be understood as enthalpies, which also implicitly partially account for the entropic component of the solvent in the implicit model. These trajectories were analyzed using the VMD program for visualization [50]. 

### 4.5. Repulsive Scaling–Replica Exchange Molecular Dynamics (RS-REMD)

#### 4.5.1. Molecular Dynamics Simulation

For molecular docking using the RS-REMD protocol introduced by Siebenmorgen et al., we applied protocols from [26] with minor modifications for protein–GAG systems [27]. The ff14SBonlysc force field parameters for endostatin [46] and the GLYCAM06 [47] for HP dp24 were used. The ligand was placed far from the N-terminus of the protein. The implicit solvent with the model igb = 8 implemented in the original approach of Siebenmorgen et al. [26] and an infinite cut-off for non-bonded interactions was used. There were 3000 steps of minimization, followed by heating to 300 K for 10 ns with a Langevin thermostat. The harmonic restraint of 0.05 kcal/(mol·Å^2^) was applied to all heavy atoms of the protein and Zn^2+^, as was the positional restraint of 1.0 kcal/(mol·Å^2^) between the center of mass (COM) of the receptor and the ligand in the production run. The 16 replicas were used, with different Lennard-Jones (LJ) parameters for atomic pairs of receptor and ligand molecules. Two parameters: d (adjusting the effective van der Waals radius, equivalent to the LJ sigma parameter) and e (changing the LJ potential well depth, equivalent to the LJ epsilon parameter) were assigned to 0.00 Å, 0.01 Å, 0.02 Å, 0.04 Å, 0.08 Å, 0.12 Å, 0.16 Å, 0.20 Å, 0.24 Å, 0.28 Å, 0.32 Å, 0.38 Å, 0.44 Å, 0.50 Å, 0.58 Å, 0.68 Å and 1.00, 0.99, 0.98, 0.97, 0.96, 0.94, 0.92, 0.90, 0.88, 0.86, 0.84, 0.82, 0.80, 0.78, 0.76, 0.74, respectively. The total duration of the production run was 60 ns per replica. The details on the protocols used by this particular approach, as well as an analysis of its prediction power for docking GAGs, are described in our previous studies [27,51,52], as well as in the original publication by Siebenmorgen et al. [26].

#### 4.5.2. Binding Free Energy Calculations/Scoring

The binding free energy was calculated using the MM-GBSA method with the model igb = 2 [53] for all 16 replicas.

#### 4.5.3. Refinement

The structures from the RS-REMD step were refined for the 20 frames with the lowest in vacuo electrostatic energy obtained from the MM-GBSA analysis. Unrestrained MD simulations 40 ns in length were performed as described above (see Molecular Dynamics and MM-GBSA Calculations Section). 

#### 4.5.4. MD Data Analysis

The RS-REMD and refinement trajectories were analyzed using the CPPTRAJ program in AMBER Tools 17 [41]. In addition, the Conformational Analysis Tools (CAT) program [38] was used for the hydrogen bond analysis, as well as for the rings’ puckering conformation analysis. The trajectories were visualized in the VMD 1.9.3 program [50], and PyMOL 1.2r3pre [54] was used for the production of figures.

### 4.6. Unbiased MD Simulation

Starting structures were built using the graphical interface of YASARA [55]. HP dp5 (GlcNS(6S)-IdoA(2S)-GlcNS(6S)-IdoA(2S)-GlcNS(6S) with IdoA(2S) in a ^1^C_4_ conformation) was positioned in the simulation box (63 Å × 63 Å × 63 Å) at a distance larger than 10 Å from the nearest protein atom. The molecular system was solvated in 0.9% NaCl solution (0.15 M) and simulations were performed at 310 K using periodic boundary conditions and the AMBER14 (ff14sb) force field [46]. GLYCAM06 [47] parameters were used for the GAGs. The box size was rescaled dynamically to maintain a water density of 0.996 g/mL. Simulations were performed using YASARA with GPU acceleration in “fast mode” (4 fs time step) [56] on “standard computing boxes” equipped with one 12-core i9 CPU and NVIDIA GeForce GTX 1080 Ti, reaching a sampling performance of about 300 ns/day. Multiple replicas were sampled, reaching accumulated timescales of about 20 μs. These simulations added up to more than 60 computing days. Snapshots were recorded every 25 ps. The stability of the complexes was monitored based on an atom–atom contact analysis (contact distance cut-off = 4 Å). CAT [38] was used for the analysis of trajectory data, general data processing and the generation of scientific plots. PyMOL [54] was used to generate molecular graphics.

## Figures and Tables

**Figure 1 molecules-29-04040-f001:**
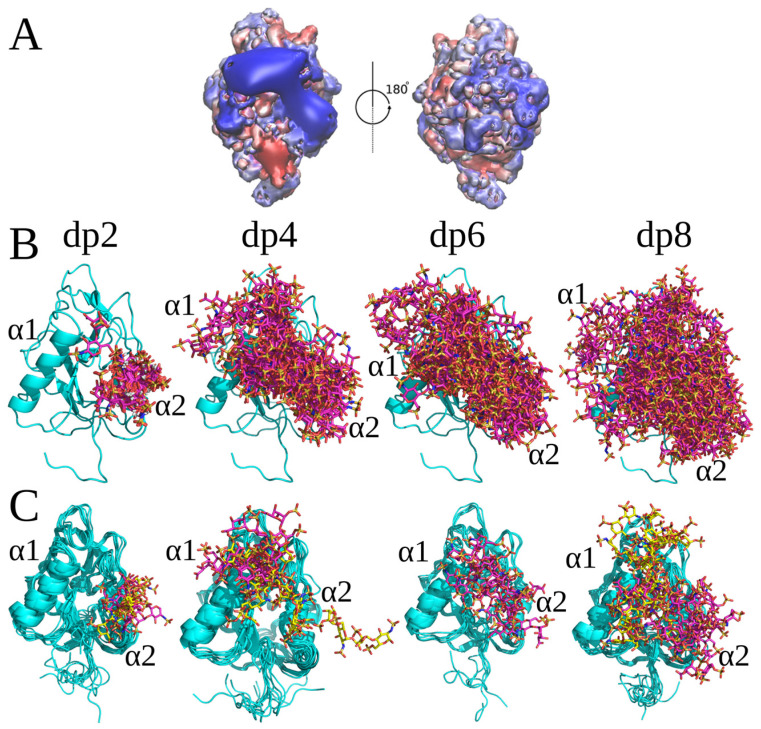
(**A**) Endostatin electrostatic potential isosurfaces (blue, 2 kcal mol^−1^ e^−1^; red, −2 kcal mol^−1^ e^−1^) (**B**) HP dp2, dp4, dp6 and dp8’s top 50 docking poses (endostatin—blue color, cartoon; HP—pink color, licorice); (**C**) structures of the complexes obtained from the last frame of MD simulations (endostatin—blue color, cartoon; HP: cluster 1—pink color, cluster 2—yellow color; licorice).

**Figure 2 molecules-29-04040-f002:**
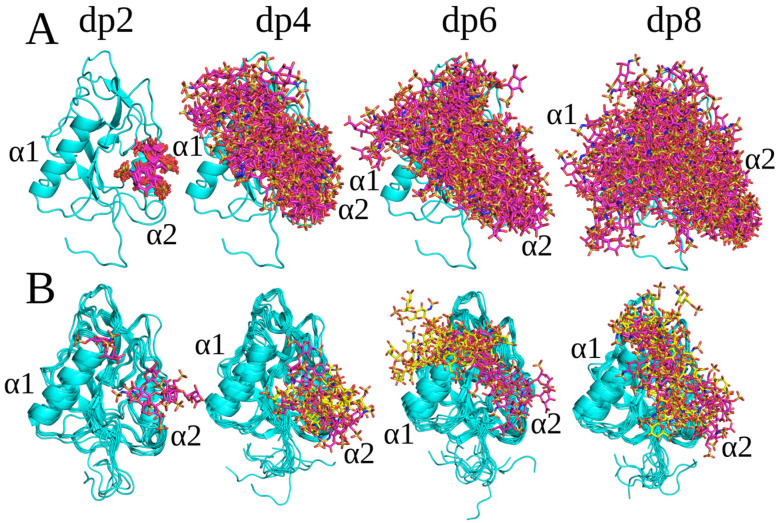
(**A**) HP ^2^S_0_ dp2, dp4, dp6 and dp8’s top 50 docking poses (endostatin—blue color, cartoon; HP—pink color, licorice) and (**B**) the structures of the complexes obtained from the last frame of MD simulations (endostatin—blue color, cartoon; HP: cluster 1—pink color, cluster 2—yellow color, licorice).

**Figure 3 molecules-29-04040-f003:**
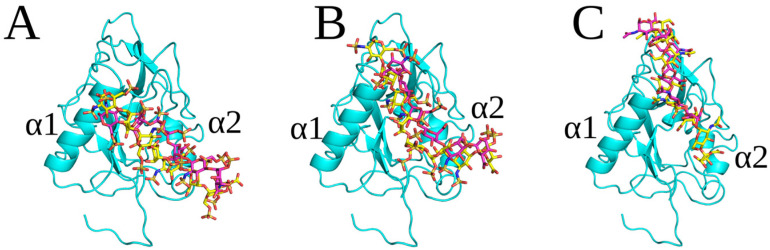
Superimposition of endostatin on the most energetically favorable complex with HP and deHS dp6 and dp8: (**A**) HP ^1^C_4_ dp6 (magenta), dp8 (yellow); (**B**) HP ^2^S_0_ dp6 (magenta), dp8 (yellow); and (**C**) desulfated HS dp6 (magenta), dp8 (yellow).

**Figure 4 molecules-29-04040-f004:**
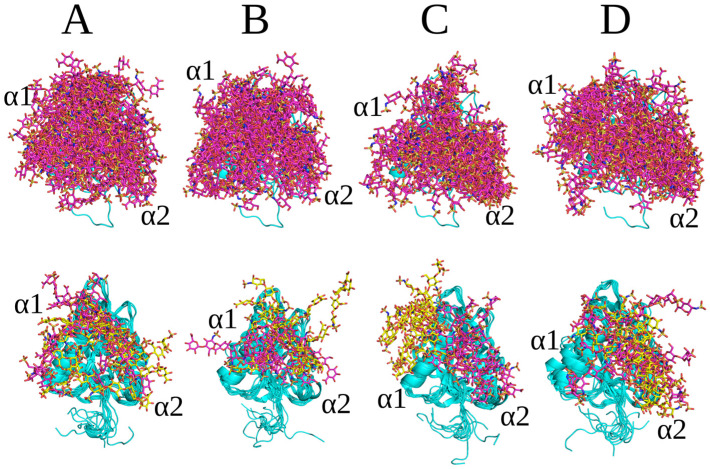
Upper panel: HS’s top 50 docking poses; lower panel: structures of the complexes obtained from the last frame of MD simulations with the repeating units (**A**) GlcNS(6S)-GlcA, (**B**) GlcNS-GlcA, (**C**) GlcNS-IdoA(2S) ^1^C_4_ and (**D**) GlcNS-IdoA(2S) ^2^S_0_ dp8 (in licorice representation, magenta—cluster 1, yellow cluster 2), with endostatin in its cartoon representation.

**Figure 5 molecules-29-04040-f005:**
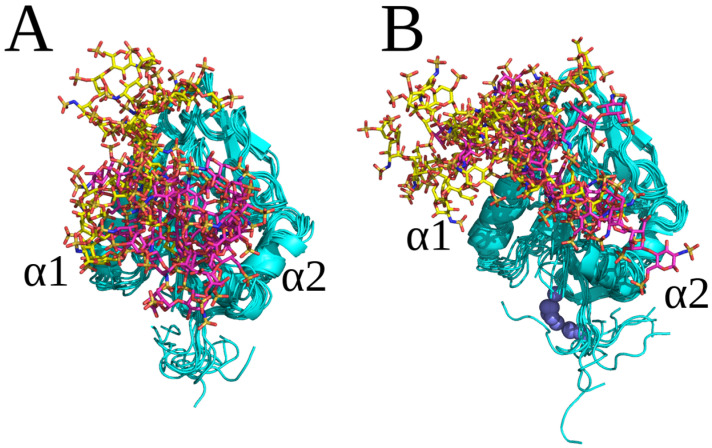
Structures of the complexes obtained from the last frame of MD simulations of endostatin–HP dp8 complexes without (**A**) and with Zn^2+^ (**B**); the protein is depicted as a cyan cartoon; Zn^2+^ is a violet sphere; and HP is presented using a licorice representation (magenta—cluster 1, yellow cluster 2).

**Figure 6 molecules-29-04040-f006:**
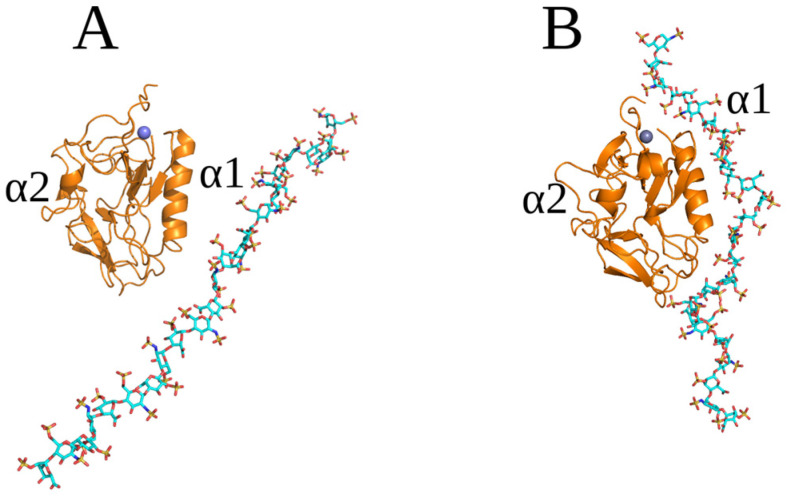
Structure of the endostatin–HP dp24 complex obtained by the RS-REMD approach (endostatin: orange cartoon, HP dp24: licorice; starting position cyan, final position green). (**A**,**B**) correspond to the final frame of the RS-REMD simulation and the final frame of the following conventional MD refinement, respectively.

**Figure 7 molecules-29-04040-f007:**
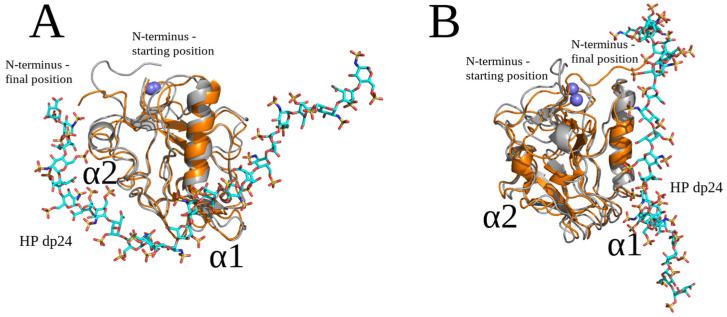
Structures of the endostatin–HP dp24 complexes after the refinement procedure for the highest-scoring pose 1 (**A**) and pose 2 (**B**); proteins are in white (starting position) and orange (final position) cartoon; Zn^2+^ is a violet sphere; and HP dp24 is in cyan licorice.

**Figure 8 molecules-29-04040-f008:**
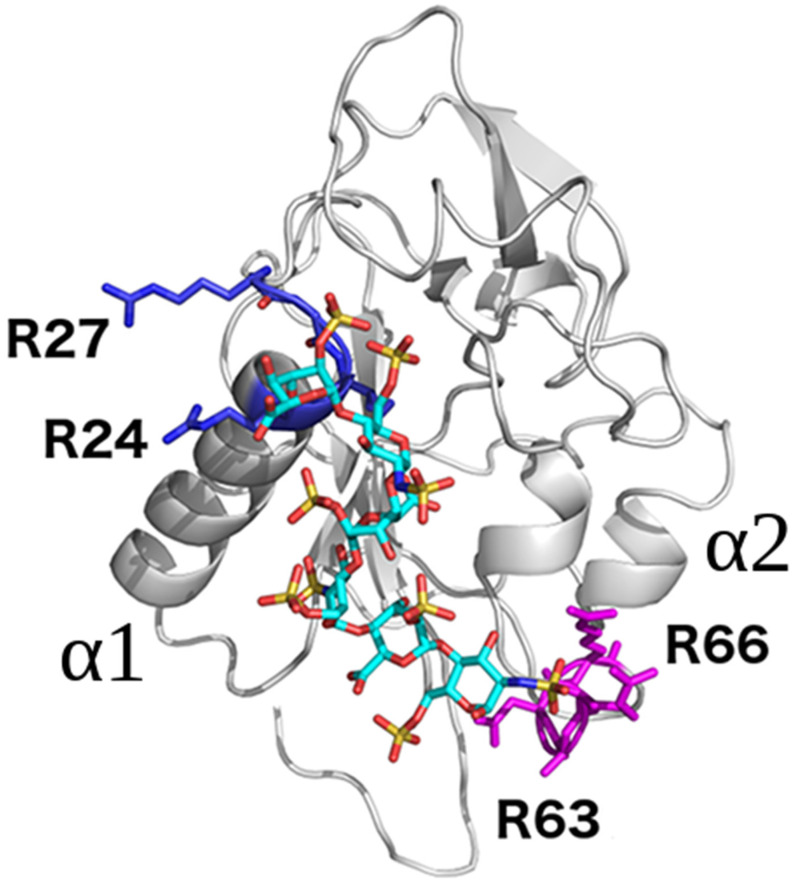
Endostatin in complex with docked HP dp8 in ribbon and cyan stick representations, respectively. The amino acids that belong to the Walker A motif RX_(2-3)_R and interact with the adenine in ATP are colored in blue and magenta, respectively.

**Figure 9 molecules-29-04040-f009:**
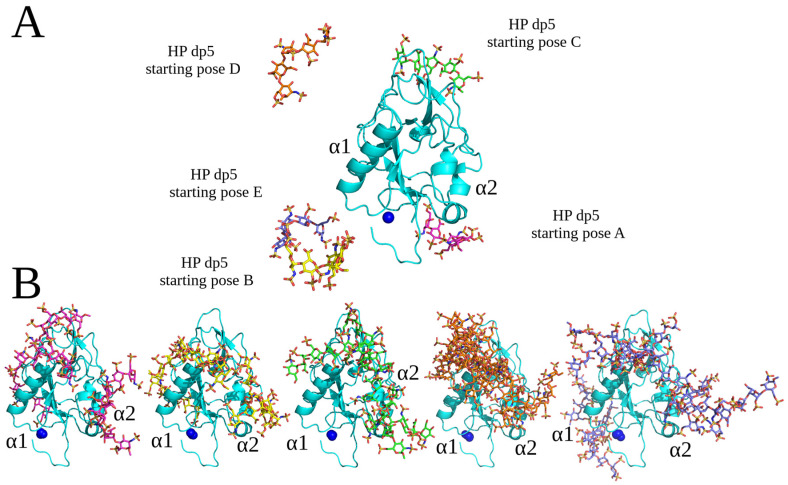
(**A**) Five starting poses of HP dp5 (endostatin—blue color, cartoon; HP—pose A pink, pose B yellow, pose C green, pose D orange and pose E purple color, licorice; Zn^2+^—blue, sphere). (**B**) Structures of the complexes obtained from the last frame of unbiased MD simulations (endostatin—blue color, cartoon; HP—pose A pink, pose B yellow, pose C green, pose D orange and pose E purple color, licorice; Zn^2+^—blue, sphere).

**Figure 10 molecules-29-04040-f010:**
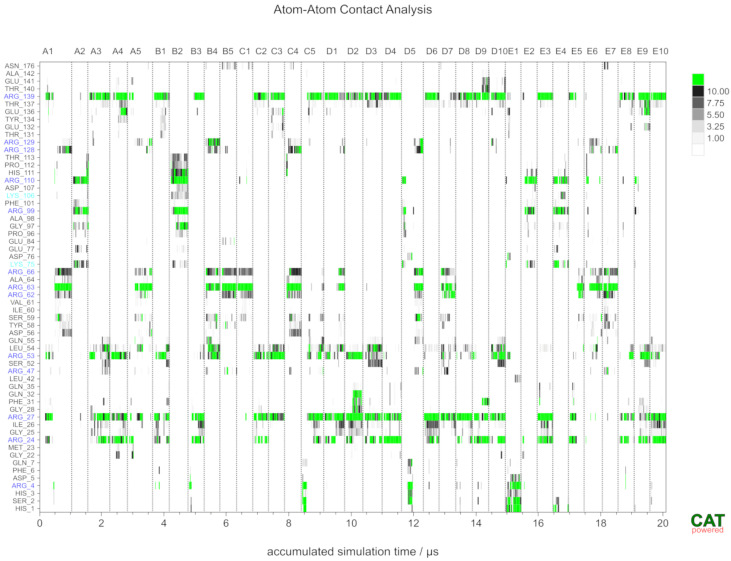
Accumulated trajectory (35 MD simulations, separated by vertical lines) of the number of atom–atom contacts between HP dp5 and endostatin residues. Only amino acids that are in contact at least 1% of the simulation time are shown. **Arg and Lys residues are highlighted in blue and cyan, respectively**.

**Table 1 molecules-29-04040-t001:** MM-GBSA binding free energies of endostatin complexes with HP and HS oligosaccharides, obtained for the whole MD trajectory.

ΔG [kcal/mol]	GAG
−30.8 ± 8.5	HP (GlcNS(6S)-IdoA(2S)) dp2, ^1^C_4_
−45.6 ± 7.7	HP (GlcNS(6S)-IdoA(2S)) dp4, ^1^C_4_
−50.3 ± 11.0	HP (GlcNS(6S)-IdoA(2S)) dp6, ^1^C_4_
−53.8 ± 17.9	HP (GlcNS(6S)-IdoA(2S)) dp8, ^1^C_4_
−33.6 ± 7.2	HP (GlcNS(6S)-IdoA(2S)) dp2, ^2^S_0_
−31.1 ± 5.7	HP (GlcNS(6S)-IdoA(2S)) dp4, ^2^S_0_
−68.5 ± 14.4	HP (GlcNS(6S)-IdoA(2S)) dp6, ^2^S_0_
−70.5 ± 14.9	HP (GlcNS(6S)-IdoA(2S)) dp8, ^2^S_0_
−18.8 ± 4.2	desulfated HS (GlcNAc-IdoA) dp2
−25.7 ± 4.2	desulfated HS (GlcNAc-IdoA) dp4
−43.4 ± 1.8	desulfated HS (GlcNAc-IdoA) dp6
−48 ± 13.6	desulfated HS (GlcNAc-IdoA) dp8
−48.2 ± 8.8	HS (GlcNS(6S)-GlcA) dp6
−47.7 ± 10.7	HS (GlcNS-GlcA) dp6
−48.0 ± 12.1	HS (GlcNS-IdoA(2S)) dp6, ^1^C_4_
−56.9 ± 11.1	HS (GlcNS-IdoA(2S)) dp6, ^2^S_0_

**Table 2 molecules-29-04040-t002:** MM-GBSA binding free energies analysis of endostatin complexes with HS dp8 in the presence of Zn^2+^.

ΔG (kcal/mol)	GAG
−62.5 ± 14.5	HP (GlcNS(6S)-IdoA(2S))_4_ ^1^C_4_
−51.8 ± 22.1	(GlcNS(6S)-GlcA)_4_
−42.9 ± 13.3	(GlcNS-GlcA)_4_
−49.2 ± 13.6	(GlcNS-IdoA(2S))_4_ ^2^S_0_
−51.9 ± 13.4	(GlcNS-IdoA(2S))_4_ ^1^C_4_

**Table 3 molecules-29-04040-t003:** MM-GBSA binding free energies of the 20 most favorable endostatin–HP dp24 complexes obtained by the RS-REMD approach after the refinement procedure.

ΔG (kcal/mol)	Complex (n°)
−91.0	1
−106.7	2
−65.7	3
−71.4	4
−80.6	5
−87.3	6
−92.0	7
−88.9	8
−56.4	9
−79.3	10
−101.1	11
−94.1	12
−78.8	13
−67.4	14
−99.8	15
−52.9	16
−105.1	17
−84.0	18
−65.7	19
−79.5	20

**Table 4 molecules-29-04040-t004:** Per-residue decomposition analysis of the 20 most favorable endostatin–HP dp24 complexes obtained by the RS-REMD approach after the refinement procedure. The ΔG values are average values for individual amino acid residues from all 20 analyzed complex structures.

ΔG [kcal/mol]	Amino Acid Residue
−3.9	His1
−4.6	Arg4
−8.0	Arg24
−12.9	Arg27
−5.9	Arg38
−5.6	Arg47
−10.8	Arg53
−5.8	Arg62
−7.8	Arg63
−4.6	Arg66
−2.8	Lys75
−2.3	Arg99
−2.9	Lys106
−2.9	Arg110
−2.4	Lys117
−3.9	Arg128
−4.5	Arg129
−11.2	Arg139
−2.4	Arg156

**Table 5 molecules-29-04040-t005:** MM-GBSA binding free energies of endostatin mutant complexes with HP dp8.

ΔG [kcal/mol]	Mutant
−79.3 ± 14.5	D30R
−81.6 ± 15.5	D56R
−64.1 ± 10.3	D65R
−107.2 ± 12.7	D30R, D56R
−111.4 ± 19.8	D30R, D56R, D65R

## Data Availability

Data are contained within the article and Appendix A.

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
