# Peer review of "Structural Insights into Endostatin–Heparan Sulfate Interactions Using Modeling Approaches"

_molecules, 2024, doi:10.3390/molecules29174040_

Round 1
Reviewer 1 Report
Comments and Suggestions for Authors
A solid theoretical work, using a variety of theoretical approaches, elucidates the mechanism of interactions between selected glycosaminoglycans (GAGs) and a fragment of the collagen XVIII molecule. My comments relate mainly to some technical issues or encourage the authors to explain some issues in more detail.
- Lines 51, 104, 194, 388: Reference errors ("+JACS GOLD here", "[Ref Mulloy]", "[Heparanome REF]", "[Refs]").
- Line 155: "Two parameters: d (adjusting the effective van der Waals radius) and e (changing the LJ potential well depths)": Are these two parameters equivalent to LJ sigma and epsilon? If yes, the usual notation should be applied. If not, their definition should be given. Moreover, if e stands for potential well depth, it should have a unit. Finally, there are 16 data points for d (which makes sense, as there are 16 replicas) but only 8 for e. Perhaps, to avoid confusion, specific combinations of (d,e) should be given for all replicas?
- The procedure described in lines 142-159 looks more like an MD protocol rather than a docking-related one. For instance, there is a time of simulation per replica and thermostat given, but no other MD-related details (e.g., integration time step). I am not familiar with the Repulsive Scaling-Replica Exchange algorithm applied for docking, and I assume that it may be true for most potential readers. I suggest elaborating more on the docking algorithm. In particular, the problem of convergence of the results should be briefly addressed.
- In Table 1, the binding energies determined for HP with the IdoA ring being in 1C4 seem quite additive with respect to dp. However, this trend is either less clear or absent in the remaining cases. What is the source of this observation? Multiple binding modes of varying magnitude from one compound to another, pure coincidence, or something else?
- I am a little bit confused about docking in the presence of the Zn2+ ions. How was the position of ions determined? Was it the subject of docking itself, in parallel to GAG docking?
- Another question with respect to docking involving Zn2+ ions: What types of interaction energies were accounted for in the energy balance when arriving at the final binding energies? In particular, how were the electrostatic energies resulting from Zn2+-protein and Zn2+-GAG interactions treated?
- I wonder if any events of ring puckering were observed during unbiased MD simulations and, if yes, if they appear to be relevant for the observed modes of GAG-protein interactions.
Reviewer 2 Report
Comments and Suggestions for Authors
The manuscript is well-written; however, there are some points that should be addressed. I hope my concerns will help the authors improve the manuscript.
Major comments:
- The authors should provide validation for the methods used in the manuscript.
- The manuscript lacks a benchmarking description for the docking section.
- The simulation time of 25 ns is insufficient and too short; the authors should increase the simulation time.
Minor comments:
- It seems that the authors forgot to include some references in the manuscript.
- The authors should explain how they selected residues for mutation. Is there any experimental evidence?
- Line 130: For producing three replicas, the authors should explain the details. How were the three replicas produced?
- Lines 143-150: The description is unclear and needs clarification.
- The description of Figure 1A is missing in the manuscript.
Comments on the Quality of English Language
The authors should correct the English errors throughout the manuscript, including grammar and spelling errors (e.g., Lines 78, 102, etc.).
Author Response
Please, see the attachment.

Round 2
Reviewer 2 Report
Comments and Suggestions for Authors
I have no further comments.